# Corticospinal-Evoked Responses from the Biceps Brachii during Arm Cycling across Multiple Power Outputs

**DOI:** 10.3390/brainsci9080205

**Published:** 2019-08-19

**Authors:** Evan J. Lockyer, Katarina Hosel, Anna P. Nippard, Duane C. Button, Kevin E. Power

**Affiliations:** 1Human Neurophysiology Lab, School of Human Kinetics and Recreation, Memorial University of Newfoundland, St. John’s, NL A1C 5S7, Canada; 2Faculty of Medicine, Memorial University of Newfoundland, St. John’s, NL A1C 5S7, Canada; 3Department of Psychology, Memorial University of Newfoundland, St. John’s, NL A1C 5S7, Canada

**Keywords:** MEP, CMEP, arm cranking, motoneurone, exercise

## Abstract

*Background*: We examined corticospinal and spinal excitability across multiple power outputs during arm cycling using a weak and strong stimulus intensity. *Methods*: We elicited motor evoked potentials (MEPs) and cervicomedullary motor evoked potentials (CMEPs) in the biceps brachii using magnetic stimulation over the motor cortex and electrical stimulation of corticospinal axons during arm cycling at six different power outputs (i.e., 25, 50, 100, 150, 200 and 250 W) and two stimulation intensities (i.e., weak vs. strong). *Results*: In general, biceps brachii MEP and CMEP amplitudes (normalized to maximal M-wave (M_max_)) followed a similar pattern of modulation with increases in cycling intensity at both stimulation strengths. Specifically, MEP and CMEP amplitudes increased up until ~150 W and ~100 W when the weak and strong stimulations were used, respectively. Further increases in cycling intensity revealed no changes on MEP or CMEP amplitudes for either stimulation strength. *Conclusions*: In general, MEPs and CMEPs changed in a similar manner, suggesting that increases and subsequent plateaus in overall excitability are likely mediated by spinal factors. Interestingly, however, MEP amplitudes were disproportionately larger than CMEP amplitudes as power output increased, despite being initially matched in amplitude, particularly with strong stimulation. This suggests that supraspinal excitability is enhanced to a larger degree than spinal excitability as the power output of arm cycling increases.

## 1. Introduction

The influence of muscle contraction intensity on the excitability of the corticospinal pathway in humans has been well-studied during isometric contractions. Most of this research has involved the use of non-invasive stimulation techniques to assess corticospinal and/or spinal excitability to muscles of the upper [1,2,3] and, to a lesser extent, the lower limb [4] across a wide range of isometric contraction intensities. In general, the findings from these studies indicate that motor evoked potentials (MEPs) elicited by transcranial magnetic stimulation (TMS) increase in size as the strength of isometric muscle contractions increase up until a peak, after which they plateau and subsequently decrease as contraction strength approaches maximal ((i.e., 100% maximal voluntary contraction (MVC)) [1,2,3,4]. This modulation in MEP is accompanied by a similar change in the cervicomedullary MEP (CMEP) elicited by transmastoid electrical stimulation (TMES) of corticospinal axons, suggesting that the change in corticospinal excitability is largely mediated by spinal factors [1,4].

Using a strong stimulus intensity (set to evoke responses equal to 65–80% the maximal compound muscle action potential (M_max_)), Martin et al. (2006) showed that MEP and CMEP areas increased linearly in size during weak isometric contractions (i.e., <50% MVC) of the biceps brachii as muscle contraction intensity increased, whereas during strong contractions (i.e., >50% MVC) MEP and CMEP areas plateaued at ~75% MVC, and subsequently decreased as the contraction intensity approached 100% MVC [1]. When a lower stimulus intensity (set to evoke responses equal to 30–50% M_max_) was used, MEP and CMEP areas followed a similar pattern of modulation with contraction intensity, however, peak responses were not observed until ~90% MVC, after which MEP and CMEP areas decreased. Moreover, the decline in MEP and CMEP area with the lower stimulus intensity was less marked than that observed when the stronger stimulus intensity was used [1]. Thus, the intensity of stimulation is an important factor to consider in assessing corticospinal excitability given how it can influence the primary measurement(s), and the associated interpretation of the data.

Substantially less information, however, is available regarding the influence of muscle contraction intensity on the modulation of corticospinal excitability during rhythmic motor outputs such as those observed during cycling [5,6,7]. This is an important topic to consider given that rhythmic motor outputs, such as arm cycling, are partially generated by spinally located networks of interneurons referred to as central pattern generators [8,9], and that corticospinal excitability is modulated differently during rhythmic locomotor outputs than during isometric contractions, indicating task-specificity [5,10,11]. In two separate studies from our lab, we have investigated changes in corticospinal and spinal excitability as arm cycling intensity (i.e., power output) was increased [6,7]. However, changes in excitability were assessed across a small range of power outputs, and thus may not have observed potential changes in excitability that occurred at higher cycling intensities. Thus, it remains unknown whether a similar peak, plateau and subsequent decline in corticospinal and spinal excitability are observed with increasing arm cycling intensity, as observed in isometric contractions.

Accordingly, the purpose of the present study was to: (1) characterize the influence of muscle contraction intensity on changes in corticospinal and spinal excitability projecting to the biceps brachii over a wide range of arm cycling intensities, and (2) assess the influence of stimulation intensity on corticospinal and spinal outputs as cycling intensity increased. Specifically, we sought to examine the effects of using a weak and a strong stimulus intensity on corticospinal and spinal excitability as power output was increased during cycling. We hypothesized that: (1) using the weak stimulus, corticospinal and spinal excitability would increase similarly across all arm cycling power outputs, and (2) using the *strong* stimulus, corticospinal and spinal excitability would increase but experience a plateau, and subsequent decrease as cycling intensity increased towards the maximum power output examined.

## 2. Materials and Methods

### 2.1. Participants

This study consisted of a familiarization session and two experimental sessions; (1) a transcranial magnetic stimulation (TMS) session and (2) a transmastoid electrical stimulation (TMES) session (see *Protocol* below). A total of nine healthy, male volunteers (24.2 ± 5.9 years, 180.7 ± 7.8 cm, 82.2 ± 8.3 kg, 1 left-hand dominant) with no known neurological impairment participated in session one, and eight of those volunteers (1 left-hand dominant) returned on a separate day (>24 h) to complete session two. In accordance with the Tri-Council guidelines in Canada, all participants gave written, informed consent prior to participating in the study, and potential risks were fully disclosed. Prior to TMS, all participants were screened for contraindications to magnetic stimulation using a safety checklist [12]. To determine limb dominance, the Edinburgh handedness inventory [13] was used. This information was gathered because all evoked responses elicited by TMS and TMES (see *Stimulation Conditions* below) were taken from the dominant arm. Additionally, all participants filled out a Physical Activity Readiness Questionnaire for Everyone (PAR-Q+, Canadian Society for Exercise Physiology (CSEP)) to screen for any contraindications to physical activity. Participants also refrained from caffeine for 12 h and alcohol for 24 h prior to each experimental session. All procedures were performed in compliance with the Declaration of Helsinki and were approved by the Interdisciplinary Committee on Ethics in Human Research (ICEHR no. 20181196-HK) at Memorial University of Newfoundland.

### 2.2. Experimental Setup

Many of the experimental procedures and recording techniques herein are similar to those described previously [6,7,14]. All sessions were conducted with participants seated upright on an arm cycle ergometer (SCIFIT ergometer, model PRO2 Total Body, Tulsa, OK, USA). The seat height of the ergometer was adjusted so that participants’ shoulders were approximately in line with the axis of rotation of the arm cranks, and the seat distance was manipulated to a position in which participants were at a comfortable distance (i.e., no reaching or trunk variation during cycling) from the hand pedals. The seat height and distance were recorded for each participant during the familiarization session and were used for the subsequent sessions. Arm cycling trials were performed in an asynchronous cranking pattern with the forearms fixed in a pronated position. Wrist braces were worn to limit the amount of wrist flexion and extension during cycling as a means to diminish the influence of short- and long-latency reflex connections that have been shown to exist between the wrist flexors and the biceps brachii (see Figure 1) [15].

For this study, participants were required to cycle at 6 different power outputs: 25, 50, 100, 150, 200, and 250 Watts (W) all at a constant cadence of 60 revolutions per minute (rpm). These cycling conditions were repeated at two different stimulation intensities (see *Stimulation Conditions* below), for a total of 12 cycling trials.

### 2.3. Electromyography Recordings

Surface electromyography (EMG) was recorded from the biceps brachii of the dominant arm using pairs of disposable Ag-AgCl surface electrodes (MediTrace^TM^ 130 Foam Electrodes with conductive adhesive hydrogel, Covidien IIC, MA, USA). Electrodes were positioned approximately 2 cm apart (center to center) over the midline of the biceps brachii and on the lateral head of the triceps brachii in a bipolar configuration. A ground electrode was positioned on the lateral epicondyle of the dominant arm. To reduce the impedance for EMG recordings, the skin was thoroughly prepared by removing hair (via a handheld razor), abraded to remove dead skin cells (via abrasive paper), and cleaned using isopropyl alcohol swabs prior to electrode placement. The EMG signals were amplified (×300, CED 1902 amplifier; Cambridge Electronic Design Ltd., Cambridge, UK), and bandpass filtered using a 3-pole Butterworth filter with cut-off frequencies of 10–1000 Hz. All analog signals were digitized at a sampling rate of 5000 Hz and stored on a laboratory computer for off-line analysis (CED 1401 interface and Signal 5.11 software; Cambridge Electronic Design Ltd., Cambridge, UK).

### 2.4. Stimulation Conditions

Recordings were made of the motor responses in the biceps brachii to three different stimulation techniques: (1) brachial plexus stimulation at Erb’s point, (2) magnetic stimulation of the motor cortex (i.e., TMS), and (3) electrical stimulation between the mastoids at the cervicomedullary junction (i.e., TMES). Motor responses were evoked during arm cycling at the 6 o’clock position, which corresponds to the mid-elbow flexion phase of arm cycling and when biceps brachii activity is relatively the largest (for a more detailed explanation of the phases of arm cycling see review by [16]). Stimulations were triggered automatically when the right hand passed a magnetic sensor on the ergometer, at either the 6 o’clock or 12 o’clock position for right-handed and left-handed participants, respectively. The intensities for all three stimulation techniques were set during arm cycling at a constant cadence of 60 rpm and power output of 25 W. For TMS and TMES, two different stimulation intensities were used: (1) a weak stimulation intensity (set to evoke responses equal to ~10% M_max_), and (2) a strong stimulation intensity (set to evoke responses equal to ~40% M_max_). These response amplitudes were chosen to provide insight into potential differences in excitability at different portions of the motoneurone pool as cycling intensity increased. All participants had prior experience with each of the stimulation procedures before participating.

### 2.5. Brachial Plexus Stimulation

For both sessions, single rectangular pulses (200-μs duration, 90–275 mA) were delivered via a DS7AH constant current stimulator (Digitimer Ltd., Welwyn Garden City, Hertfordshire, UK) to the brachial plexus at Erb’s point to elicit maximal compound muscle action potentials (maximal M-wave (M_max_)) in the biceps brachii. The cathode was placed in the supraclavicular fossa and the anode over the acromion process. Stimulus intensity was initially set at 25 mA and was gradually increased until the size of the M-wave plateaued (i.e., M_max_). At this point, the stimulation intensity was increased by 10% (supramaximal) to ensure that M_max_ was elicited throughout the remainder of the study.

### 2.6. TMS

TMS was delivered over the vertex of the motor cortex to elicit MEPs in the biceps brachii using a Magstim 200^2^ magnetic stimulator (Magstim, Whitland, Dyfed, UK) and circular coil (13.5 cm outside diameter). The vertex was measured and marked on the participant’s scalp with a felt-tip permanent marker. One investigator ensured proper and consistent coil placement directly over vertex throughout the experiment. The coil was held firmly against the participant’s skull, parallel to the floor with the direction of current flow-oriented to preferentially activate either the left or right motor cortex, depending on hand dominance (i.e., “A” side up for right-handed participants, “B” side up for left-handed participants). Initially, TMS intensity was set at 25% of maximal stimulator output (MSO) and was increased until MEPs were observed in the biceps brachii equal in amplitude to ~10% M_max_. Once found, a trial consisting of 8 TMS was performed to ensure that the average MEPs were ~10% M_max_. This stimulation intensity was recorded as the weak stimulation intensity, and was then used for the remainder of the experiment. For the strong stimulation intensity, the same procedures were performed except the %MSO was increased until MEPs from the biceps brachii were equal in amplitude to ~40% M_max_. Once again, a trial consisting of 8 TMS was performed to ensure that the intensity of TMS would evoke MEPs equal to ~40% M_max_. Once determined, this intensity was recorded and then used as the strong intensity for the rest of the experiment.

### 2.7. TMES

TMES was delivered (200 μs pulse-width duration, DS7AH, Digitimer Ltd., Welwyn Garden City, Hertfordshire, UK) to the corticospinal axons at the cervicomedullary junction to elicit CMEPs in the dominant arm biceps brachii. Self-adhesive Ag-AgCl surface electrodes were placed on the skin at the grooves between the mastoid processes and the occipital bone, with the anode and cathode on the side corresponding to each participant’s dominant and non-dominant arm, respectively. Similar to the procedures for setting the stimulation intensities for TMS (see *TMS* above), the intensity of electrical stimulation was gradually increased (initially from 25 mA) until the amplitudes of the CMEPs were equal in amplitude to ~10% M_max_ (for the weak stimulation intensity) and ~40% M_max_ (for the strong stimulation intensity). Trials of 8 CMEPs were evoked at each stimulation intensity and the average was calculated. These stimulation intensities were recorded, and were then used for the remainder of the experiment. The latency of responses was monitored carefully to ensure that stimulation did not activate the corticospinal axons at or near the ventral roots, which would be indicated by a reduction in latency by ~2 ms [17,18].

### 2.8. Protocol

Following familiarization, participants were randomly assigned to complete either session one (TMS) or session two (TMES) first. For both sessions, the procedures were identical with the exception of the stimulation type. Following EMG preparation and ergometer modifications, stimulation intensities were determined (see above). In both sessions, M_max_ was determined first followed by the setting of stimulation intensities for the weak and strong stimulations for either TMS (session one) or TMES (session two). Once stimulation intensities were determined, participants began the 12 cycling trials consisting of six power outputs (25, 50, 100, 150, 200, and 250 W) performed at a constant cadence of 60 rpm with either the weak or strong stimulation intensity (i.e., six cycling trials at each stimulation intensity). The order of the cycling trials was randomized for each participant. While cycling, as the dominant hand passed the 6 o’clock position, one M_max_ and either six MEPs or six CMEPs (depending on the session) were evoked in a randomized order. The time between stimulations was 5–6 s. The total length of each trial was approximately 30 s. To reduce the potential influence of fatigue, one-minute rest periods were given following completion of the lower power output trials (i.e., 25, 50, 100 W), and two-minute rest periods were given after the higher power output trials (i.e., 150, 200, 250 W). Additionally, half-way through the 12 trials (i.e., after trial six), a 5-min rest period was given before the remainder of the trials were completed.

### 2.9. Data Analysis

For analysis of M_max_, MEP, and CMEP, the averaged peak-to-peak amplitudes from each cycling trial were measured from the biceps brachii of the dominant arm. Since M_max_ is thought to represent the maximal response of the motor system [4], averaged MEPs (*n* = 6) and CMEPs (*n* = 6) from each trial were normalized to the M_max_ within each cycling trial. Response latencies of all evoked responses were carefully monitored throughout all cycling trials as well. The latency for each response was classified as the duration from the stimulus artifact to the initial deflection in the voltage trace from baseline and was averaged across the total number of stimulation trials. Additionally, since the level of voluntary muscle contraction could potentially have an influence on changes in MEP and CMEP amplitudes, pre-stimulus EMG was measured from the rectified virtual channel created for the biceps and triceps brachii as the mean of a 50 ms window immediately prior to the stimulation artifact [14]. For two participants who completed CMEPs (*n* = 8), pre-stimulus EMG from the triceps brachii was not available due to a technical error during data collection. Therefore, the final sample size for CMEP pre-stimulus EMG data from the triceps brachii was *n* = 6.

### 2.10. Statistical Analysis

Group data are presented as means ± SD in the text and means ± SE in the figures (with *n* in the legends). All statistics were performed using IBM’s SPSS Statistics (IBM SPSS Statistics for Windows, Version 23.0. Armonk, NY, USA: IBM Corp.). Mauchly’s test was employed to assess the assumption of sphericity for repeated measures analysis. In cases where sphericity was violated, the appropriate correction was applied (i.e., Greenhouse Geisser or Huynh-Feldt) and the degrees of freedom were adjusted. Separate two-way repeated-measures ANOVAs were used to assess the effects of stimulation intensity and cycling intensity (and any interaction) on the M_max_, MEP, and CMEP amplitudes (both normalized to M_max_), the average pre-stimulus EMG, and the MEP/CMEP ratios. Post hoc pairwise comparisons were made between means using the Bonferroni correction. Additionally, because one of our aims was to examine the effects of cycling intensity on corticospinal excitability measures within each stimulation intensity (weak and strong), separate one-way repeated-measures ANOVAs were conducted for both the weak and strong stimulus on M_max_, MEP, and CMEP amplitudes (normalized to M_max_), pre-stimulus EMG, and MEP/CMEP ratios as cycling intensity increased. If a main effect was identified, post hoc pairwise comparisons were made between means using the Bonferroni correction. Independent samples *t*-tests were conducted to compare whether MEPs and CMEPs (normalized to M_max_) at both stimulation intensities were matched appropriately. To compare between MEP and CMEP amplitudes (normalized to M_max_) at each power output, independent sample *t*-tests were used with a Bonferroni correction. Paired samples *t*-tests were conducted on MEP/CMEP ratios between stimulation strengths (weak vs. strong) at each power output. All statistics were performed on group data and statistical significance was set at *p* < 0.05.

## 3. Results

Evoked responses (i.e., M_max_, MEPs, and CMEPs) were recorded from the dominant arm biceps brachii at two different stimulation intensities while participants performed arm cycling bouts over a range of contraction strengths. MEPs and CMEPs (normalized to M_max_) were evoked on separate days but were initially matched to equal 10% (weak stimulus) and 40% (strong stimulus) of the M_max_ on each day. MEPs and CMEPs were not significantly different when either the weak or the strong stimulation intensity were examined (*p* > 0.05 for both conditions), suggesting that the responses were indeed matched initially between days.

### 3.1. Biceps Brachii Evoked Responses

#### 3.1.1. MEP Amplitude

Figure 2 (top panel) and Figure 3A show representative and grouped data, respectively for MEP amplitudes from the biceps brachii during arm cycling across the various contraction intensities. Figure 2 shows evoked potential traces from one participant during arm cycling with the weak stimulation intensity. In this example, the amplitudes of the MEPs show a progressive and generally consistent increase from the lowest (25 W) to the highest (250 W) arm cycling/muscle contraction intensity. Results from the two-way ANOVA on MEP amplitudes showed a significant main effect for both stimulation intensity (strong > weak, *F*_5,40_ = 96.81, *p* < 0.001) and cycling intensity (*F*_1,8_ = 65.30, *p* < 0.001). Bonferroni post hoc tests revealed that MEP amplitudes at 25 W and 50 W were not different from one another (*p* = 0.187) but were significantly smaller than MEP amplitudes evoked during the 100, 150, 200, and 250 W trials (*p* < 0.05 for all comparisons). Additionally, there was a significant interaction between the intensity of stimulation and the intensity of cycling on MEP amplitudes (*F*_5,40_ = 65.30, *p* < 0.001). Further analysis, through use of one-way ANOVAs for each stimulation intensity, showed a significant main effect for cycling intensity on MEP amplitudes at both the weak (*F*_5,40_ = 55.61, *p* < 0.001) and strong (*F*_5,40_ = 41.28, *p* < 0.001) stimulation conditions. Using the weak stimulation, Bonferroni post hoc tests revealed that MEP amplitudes increased as cycling intensity increased up until 200 W (200 W > 150 W >100 W > 50 W > 25 W, *p* < 0.05 for all comparisons) after which MEPs plateaued (*p* > 0.05). Using the strong stimulation, MEP amplitudes similarly increased with cycling intensity, however, a peak was observed at 100 W (100 W > 50 W > 25 W, *p* < 0.05 for all comparisons), at a lower power output than that observed using the weaker stimulation condition (i.e., 200 W). Beyond 100 W, there were no further increases in MEP amplitudes (*p* > 0.05).

#### 3.1.2. Biceps Brachii Pre-stimulus EMG

Figure 3C shows group data for biceps brachii pre-stimulus EMG prior to MEPs during arm cycling. Results from the two-way ANOVA showed that mean biceps brachii pre-stimulus EMG in the 50 ms preceding an MEP was not different between the weak and strong stimulation intensity (*F*_1,8_ = 1.42, *p =* 0.267). Therefore, the average pre-stimulus EMG was pooled between the weak and strong stimulation conditions, which are represented in Figure 3C. There was a significant main effect on biceps brachii pre-stimulus EMG for cycling intensity (*F*_1.76,14.12_ = 29.33, *p* < 0.001), but, there was no interaction between stimulation intensity and cycling intensity (*F*_1.96,27.35_ = 1.96, *p* = 0.137). To further examine changes in pre-stimulus EMG with cycling intensity, one-way ANOVAs were performed. Pre-stimulus EMG increased as cycling intensity increased up until 200 W (Figure 3C, *p* < 0.05), and no differences were observed between the 200 W and 250 W conditions (*p* = 1.00).

#### 3.1.3. Triceps Brachii Pre-Stimulus EMG

Figure 3E shows group data for triceps brachii pre-stimulus EMG prior to MEPs. Similar to the biceps, results from the ANOVA showed no effect of stimulation intensity on triceps brachii EMG activity prior to a MEP (Figure 3E, *F*_1,8_ = 0.100, *p* = 0.760), but there was a significant main effect of cycling intensity (*F*_1.62,12.94_ = 19.32, *p* < 0.001). Also, there was no significant interaction between cycling intensity and stimulation intensity (*F*_5,40_ = 0.803, *p* = 0.554). To further examine the effect of cycling intensity on triceps brachii pre-stimulus EMG, one-way ANOVAs were performed using the pooled data. Results from these tests indicated that as cycling intensity increased, triceps brachii pre-stimulus EMG values were only significantly different at 150 W and 200 W. Specifically, triceps brachii pre-stimulus EMG was larger at 150 W than at 100 W (*p* = 0.006) and was larger at 200 W than 150 W (*p* = 0.044).

#### 3.1.4. CMEP Amplitude

Figure 2 (middle panel) and Figure 3B show representative and grouped data, respectively for CMEP amplitudes during the arm cycling bouts. Figure 2 portrays data from one participant from the weak stimulation intensity condition. Similar to the MEP amplitudes, in this example, CMEP amplitudes increase in a relatively consistent and progressive manner. The results from the two-way ANOVA on CMEP amplitudes showed significant main effects for both stimulation intensity (strong > weak, *F*_1,7_ = 91.50, *p* < 0.001) and cycling intensity (*F*_3.81,26.65_ = 20.16, *p* < 0.001), however, there was no significant interaction between the two factors (*F*_5,35_ = 1.34, *p* = 0.271). For cycling intensity, Bonferroni post hoc analysis revealed that CMEPs at 25 and 50 W are smaller than those at all other cycling intensities (i.e., 100, 150, 200, and 250 W) (*p* < 0.05 for all comparisons). To decipher specific effects of cycling intensity within each stimulation condition, separate one-way ANOVAs for the weak and strong stimulation conditions were performed on CMEP amplitudes. The results from the one-way ANOVAs showed a significant main effect for cycling intensity on CMEP amplitudes at both the weak (*F*_5,35_ = 21.11, *p* < 0.001) and strong (*F*_5,35_ = 9.95, *p* < 0.001) stimulation conditions. For the weak stimulation condition, Bonferroni post hoc analyses revealed that CMEP amplitudes increased up until 150 W (150 W > 100 W > 50 W > 25 W; *p* < 0.05 for all comparisons), after which CMEP amplitudes did not change (*p* > 0.05). When the strong stimulation intensity was used, post hoc analyses revealed that CMEP amplitudes increased up until 100 W (100 W > 50 W > 25 W, *p* < 0.05 for all comparisons), after which CMEPs plateaued (*p* > 0.05).

#### 3.1.5. Biceps Brachii Pre-Stimulus EMG

Figure 3D shows group data for biceps brachii pre-stimulus EMG prior to CMEPs during arm cycling. Results from the two-way ANOVA showed that mean biceps brachii pre-stimulus EMG in the 50 ms preceding CMEPs was not influenced by stimulation intensity (*F*_1,7_ = 0.02, *p* = 0.906), thus the data was pooled between the weak and strong stimulation conditions as shown in Figure 3D. There was a significant main effect on biceps brachii pre-stimulus EMG for cycling intensity (*F*_1.49,10.41_ = 43.08, *p* < 0.001), but, there was no interaction between stimulation intensity and cycling intensity (*F*_5,35_ = 1.22, *p* = 0.320). To further examine changes in pre-stimulus EMG with cycling intensity, one-way ANOVAs were performed using the pooled data. Similar to MEPs, pre-stimulus EMG for CMEPs increased as cycling intensity increased up until 200 W (Figure 3D, *p* < 0.05), and there was no difference between the 200 W and 250 W conditions (*p* = 0.885).

#### 3.1.6. Triceps Brachii Pre-Stimulus EMG

Figure 3F shows group data for triceps brachii pre-stimulus EMG prior to CMEPs. Similar to above, results from the two-way ANOVA showed no effect of stimulation intensity (*F*_1,5_ = 0.761, *p* = 0.423) and thus, the data was pooled between the week and strong stimulation intensities (Figure 3F). There was, however, a significant main effect of cycling intensity (*F*_1.31,6.55_ = 14.04, *p* = 0.006) on triceps brachii pre-stimulus EMG, but no significant interaction (*F*_5,25_ = 0.961, *p* = 0.460). To further examine the effect of cycling intensity on triceps brachii pre-stimulus EMG, one-way ANOVAs were performed using the pooled data. Results from these tests indicated that triceps brachii pre-stimulus EMG values for CMEPs were only increased at 150 W, 200 W and 250 W compared to the 25 W condition (*p* < 0.05 for all comparisons). However, triceps brachii pre-stimulus EMG was not significantly different with increased cycling intensity from 150 W to 250 W (*p* > 0.05 for all comparisons).

#### 3.1.7. MEP/CMEP Ratios

Although MEPs and CMEPs were evoked on separate days, the responses were initially matched in amplitude to approximately 10% or 40% M_max_ for the weak and strong stimulation conditions, respectively (*p* > 0.05 for both stimulation conditions). Thus, MEP amplitudes were expressed relative to CMEP amplitudes and multiplied by 100% to obtain MEP/CMEP percentages for each participant (Figure 4). This was done in an attempt to isolate whether changes in overall excitability could be attributed to changes in supraspinal and/or spinal excitability. Values greater than 100% indicate that MEP amplitudes are larger than CMEP amplitudes, suggesting that supraspinal excitability may be increased. Similarly, values less than 100% indicate that MEP amplitudes are less than CMEP amplitudes, suggesting that changes in spinal excitability are important factors in maintaining excitability of the corticospinal pathway. Results from the two-way ANOVA revealed a significant main effect for stimulation intensity (weak > strong, *F*_1,7_ = 6.94, *p* = 0.034) and cycling intensity (*F*_5, 35_ = 9.71, *p* < 0.001). Bonferroni post hoc tests revealed that MEP/CMEP at 25 W and 50 W were not different from one another (*p* = 0.413) but were significantly smaller than MEP/CMEP at 100, 150, 200, and 250 W trials (*p* < 0.05 for all comparisons). As well, there was a significant interaction effect (*F*_5, 35_ = 8.18, *p* < 0.001) between stimulation intensity and cycling intensity on MEP/CMEP ratios. To examine changes in MEP/CMEP with increased power output, one-way ANOVAs were conducted within each stimulation intensity. Results from the one-way ANOVAs showed a significant main effect for cycling intensity on MEP/CMEP ratios at both the weak (*F*_5,35_ = 9.44, *p* < 0.001) and strong (*F*_5,35_ = 4.60, *p* = 0.003) stimulation conditions. When the weak stimulation was used, Bonferroni post hoc analysis revealed that MEP/CMEP were only significantly larger than that at 25 W at 150 W (*p* = 0.037), and 200 W (*p* = 0.05). When the strong stimulation intensity was used, MEP/CMEP were significantly larger at 50 W than at 25 W (*p* = 0.026) but were not different for any other comparison. To compare changes in MEP/CMEP between the weak and strong stimulation intensities, paired samples *t*-tests were performed at each power output. Thus, a total of six comparisons were made. The *t*-tests revealed that the MEP/CMEP ratios were not significantly different at 25 W (*t*_(7)_ = 1.22, *p* = 0.261) or 50 W (*t*_(7)_ = 0.52, *p* = 0.622) when either the weak or strong stimulus was used. However, MEP/CMEP ratios were significantly larger at 100 W (*t*_(7)_ = 2.51, *p* = 0.041), 150 W (*t*_(7)_ = 3.24, *p* = 0.014), 200 W (*t*_(7)_ = 3.03, *p* = 0.019), and 250 W (*t*_(7)_ = 2.41, *p* = 0.047) when the weak stimulation was used.

#### 3.1.8. M_max_ Amplitude

For both the TMS and TMES sessions, the results from the two-way ANOVA revealed similar effects on biceps brachii M_max_ amplitudes. For both sessions, there was no effect of stimulation intensity (TMS: *F*_1,8_ = 0.093, *p* = 0.769, TMES: *F*_1,7_ = 1.06, *p* = 0.337), but there was a significant main effect for cycling intensity (TMS: *F*_5,40_ = 15.66, *p* < 0.001; TMES: *F*_1,7_ = 8.89, *p* < 0.001) on M_max_ amplitudes (Figure 5). As cycling intensity increased M_max_ amplitudes decreased (Figure 5A,B). Additionally, there was no interaction observed between factors on either day (TMS: *F*_5,40_ = 0.836, *p* = 0.532, TMES: *F*_5,35_ = 0.430, *p* = 0.825). Since there was no effect of stimulation intensity on M_max_ values, the averages from each stimulation condition (weak and strong) were pooled across the cycling intensities for each session (as shown in Figure 5). For cycling intensity, Bonferroni post hoc analyses indicated that M_max_ values decreased for the TMS and TMES session as cycling intensity increased from 25 to 250 W.

## 4. Discussion

This study shows that the amplitudes of TMS-evoked MEPs and TMES-evoked CMEPs increase with power output and plateau, but do not decrease in amplitude as has been previously shown by others during intense tonic contractions [1,4]. MEP amplitudes were much larger than CMEP amplitudes as power output increased regardless of stimulation strength, despite being initially matched in amplitude (Figure 3A,B and Figure 4). This finding suggests that supraspinal factors mediate the change in overall corticospinal excitability observed during arm cycling as intensity increases. Importantly, stimulus strength had a substantial effect on MEP and CMEP amplitudes as cycling power output increased. Responses evoked by the weak stimulation (10% M_max_) increased up to approximately 200 W for MEPs (Figure 3A and Figure 4) and 150 W for CMEPs (Figure 3B and Figure 4), whereas with the strong stimulation (40% M_max_), responses reached a peak at 100 W for both MEPs and CMEPs and did not change afterward. Thus, the MEP/CMEP ratio used as a measure of supraspinal excitability was influenced by stimulation strength, which would lead to different conclusions on mechanisms of enhanced corticospinal excitability during arm cycling as power output increases.

### 4.1. Modulation of Corticospinal and Spinal Excitability with Cycling Intensity

Past research involving isometric contractions has shown that biceps brachii MEPs and CMEPs increase up until a peak at ~75–90% MVC [1,2,3], a finding which has been attributed to the motor unit firing and recruitment characteristics of the biceps brachii during progressively stronger isometric contractions [19,20]. Following the peak, there is a subsequent decline in responses as contraction intensity approaches 100% MVC [1] which is thought to reflect the inability for some motoneurones to fire in response to artificial excitatory input at strong contraction strengths, given the high degree of voluntary input to the motoneurone pool and the associated changes in their intrinsic properties [1]. In the present study, we did not observe a decline in corticospinal excitability as arm cycling intensity increased to the maximum intensity employed. Instead, we observed a plateauing of responses for both MEPs and CMEPs at intensities below 250 W, which were differentially influenced by stimulus strength (Figure 3A,B). Our results, however, do coincide with findings from the only other study to examine corticospinal excitability changes during a locomotor-like output over a wide range of contraction intensities [5]. In that study, MEPs and CMEPs from the knee extensors during leg cycling increased in amplitude up to 300 W, after which there was a plateauing, but no decline as cycling intensity increased to 400 W [5]. Taken together, these studies suggest task-dependent changes in corticospinal and spinal excitability may be present, a finding we have previously reported [10,16,21].

In the current study, MEP and CMEP amplitudes increased at the lower, but not higher power outputs (Figure 3A,B), suggesting that the increase in overall corticospinal excitability at the low intensities (i.e., 25 to 100 W) is partially generated by increased spinal excitability. These finding are partially supported by biceps brachii pre-stimulus EMG values which increase for both stimulation types (Figure 3C,D) at the low cycling intensities, but are not significantly different between the highest cycling intensities (200 and 250 W). While this may explain the enhanced spinal excitability at the low power outputs, it does not explain why we observed a plateau in CMEP amplitudes beyond 150 W for the weak stimulus and 100 W for the strong stimulus in the present study, since EMG was still increasing beyond these power outputs. It is noted, however, that Weavil and colleagues showed increased EMG and workloads without changes in MEP and CMEP amplitudes. During isometric contractions, the biceps brachii is capable of recruiting additional motor units during contractions up to and beyond 90% MVC [19,20], which help to explain why CMEPs continue to increase beyond 90% MVC [1]. Corticospinal excitability to the biceps brachii is also task- [10,16] and forearm position-dependent [21] which is an important consideration when a comparison to tonic contractions is made. However, the lack of increase in CMEP amplitudes beyond 150 W and 100 W during arm cycling in the current study, while MEPs and background EMG are still increasing is unlikely to be explained by reaching the maximum motor unit recruitment of the biceps, given that these cycling intensities are not maximal, at least relative to a sprint test [7]. It is possible, however, that motoneurone recruitment strategies during a rhythmic motor output such as arm cycling may be different from those observed during isometric contractions (Power et al., 2018), and therefore could cause motoneurones to be maximally recruited sooner than 90% of maximal cycling power. Work in adult decerebrate cats and rats, for example, demonstrated that spinal motoneurones are characterized by changes in their electrical properties during locomotor outputs that would act to enhance their recruitment and firing [22,23,24]. These same changes in motoneurone excitability do not occur during tonic motor output [23].

### 4.2. Modulation of Supraspinal Excitability with Cycling Intensity

In the current study, MEP/CMEP ratios increased with power output, in particular when the weak stimulation intensity was used (Figure 4) suggesting that supraspinal excitability was enhanced to a larger degree than spinal excitability. It is plausible that changes in the excitability of interneuronal circuits and/or interhemispheric connections may be involved. During tonic contractions, short-interval intracortical inhibition (SICI) is reduced as muscle contraction intensity increases [25,26,27], a finding that is thought to downregulate the activity of the inhibitory neurons which project onto corticospinal cells involved in producing the movement. We recently showed that SICI was present during arm cycling, albeit not different than a tonic contraction [28]. Thus, it is possible that reductions in SICI during arm cycling as power output increases may underlay increases in MEP amplitudes as has been shown during tonic contractions.

Another potential mechanism involves cortical spread from the non-dominant to the dominant motor cortex as we have previously hypothesized [6,7,14]. Since arm cycling is a bilateral motor output it is possible that cortical excitation arising from the active, non-dominant motor cortex could facilitate excitability in the dominant motor cortex, which could reduce the input required to induce an MEP by a given TMS pulse. However, when the strong stimulation intensity was used, the changes in MEP/CMEP ratios were less marked and did not increase as cycling intensity increased suggesting a ceiling effect in the MEP amplitudes had been reached.

### 4.3. Differences between Stimulation Intensities

This study highlights the importance of stimulation intensity selection for experimental design during locomotor outputs. Notably, MEPs continued to increase with cycling intensity up until approximately 200 W when elicited with weak stimulation intensity (10% M_max_), while they plateaued at approximately 100 W under strong (40% M_max_) stimulation. This led us to conclude that supraspinal excitability increases with increased power output, an effect only observed when weak stimulus intensity was used. In contrast, using the strong stimulation intensity leads one to believe, perhaps falsely, that spinal factors were driving the change in overall corticospinal excitability as a function of power output, a conclusion also reached by Weavil and colleagues (2015) who used a strong stimulation intensity (MEPs and CMEPs were ~50% M_max_). The use of a weak stimulation intensity yielded a more precise measure of corticospinal excitability in this specific study as MEPs were less susceptible to ceiling effects than at the strong stimulation.

### 4.4. Methodological Considerations

An important methodological consideration in interpreting the current data is that we did not make the power outputs relative to each individual as we have recently done in two separate studies during arm cycling [6,7]. In Spence et al. (2016) we used 5 and 15% of peak power output determined by a sprint test (modified Wingate) while in Lockyer et al. (2018) we used 20, 40, and 60% of peak power output determined via a standard incremental aerobic test (20 W increases every two minutes) [29]. These methods were not without limitations, however. The former used a sprint test to prescribe aerobic cycling intensity at 60 RPM and the latter incremental test resulted in most of the participants reaching a similar peak power output of ~120 W. In the present study we used absolute power outputs as has been used by others [5,30] and all participants were able to cycle well above the aerobic test maximum power output of 120 W obtained in our prior work. We were thus able to have participants cycle at supramaximal intensities, albeit we did not quantify exertion levels. Additionally, the sample size of (*n* = 9) for MEPs and (*n* = 8) for CMEPs was not determined by a power analysis and therefore, it is unclear whether a larger sample size would have influenced the present results.

## 5. Conclusions

The present study describes the influence of stimulation strength over a wide range of cycling intensities on corticospinal and spinal excitability during arm cycling. We have demonstrated that corticospinal excitability to the biceps brachii is increased with cycling intensity during low power outputs, a finding that is partially mediated by spinal factors. As cycling intensity increases, however, it appears as though supraspinal factors may play more of a role in modulating overall corticospinal excitability. Additionally, this study highlights the importance of stimulation intensity selection to assess corticospinal excitability during motor output. It is concluded that the use of a weaker stimulation intensity provides a more precise measure of corticospinal excitability during locomotor outputs at high intensities as they are less susceptible to potential ceiling effects.

## Figures and Tables

**Figure 1 brainsci-09-00205-f001:**
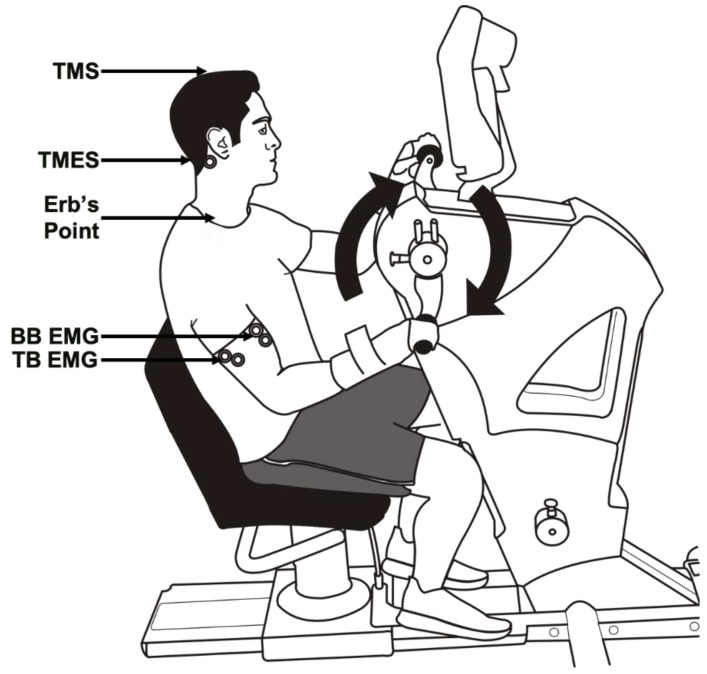
Experimental setup for arm cycling trials showing participant seated on the ergometer instrumented with surface EMG electrodes on the biceps and triceps brachii. Arrows point to the site of each stimulation technique. All arm cycling trials were conducted in the forward direction. Abbreviations: TMS, transcranial magnetic stimulation; TMES, transmastoid electrical stimulation; BB, biceps brachii; TB, triceps brachii; EMG, electromyography.

**Figure 2 brainsci-09-00205-f002:**
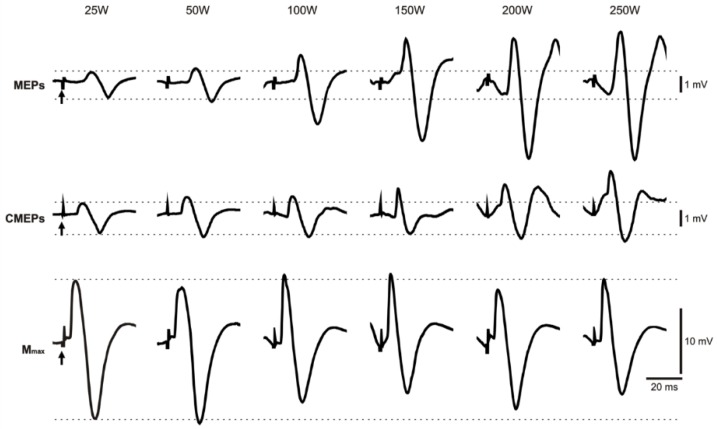
Raw traces for MEPs (top row), CMEPs (middle row), and M_max_ (bottom row) from the biceps brachii of a single participant (*n* = 1) across arm cycling power outputs using the weak stimulation intensity. Each MEP and CMEP waveform represent the average of six evoked potentials. Arrows indicate the stimulus artifact, and dashed lines portray the initial amplitudes of evoked potentials with the *weak* stimulation (~10% M_max_). In this example, MEP and CMEP amplitudes show a general progressive increase as power output increases towards 250 W, while M_max_ gradually decreases.

**Figure 3 brainsci-09-00205-f003:**
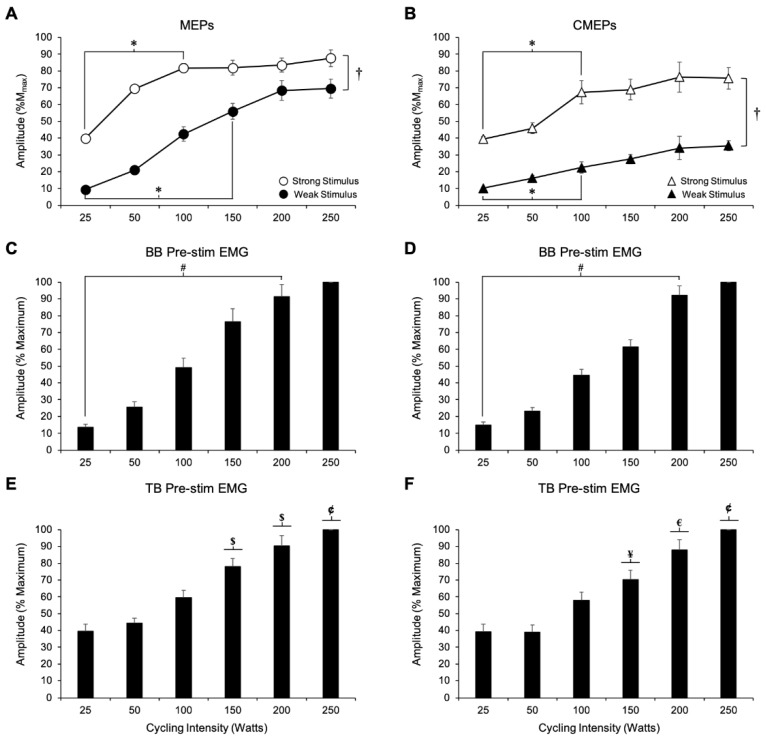
(**A**,**B**) Normalized grouped data (means ± SE) of the peak-to-peak amplitudes for MEPs (**A**) and CMEPs (**B**) obtained from the biceps brachii at each power output examined. MEPs and CMEPs were normalized to M_max_ at each corresponding cycling intensity. In both A and B, filled data points represent when the weak stimulus was used, while unfilled points represent data from the strong stimulus. For clarity, circles were used for MEPs, while triangles were used for CMEPs. In some cases, data points are bigger than SE bars. * Significant difference between illustrated data points. ^†^ Significant main effect for stimulation strength (*p* < 0.05). (**C**,**D**) Pre-stimulus EMG (means ± SE) from the biceps brachii which has been pooled and averaged between both stimulation intensities for the TMS session (**C**) and TMES session (**D**), respectively. ^#^ Significant difference between all data points. (**E**,**F**) Pre-stimulus EMG (means ± SE) from the triceps brachii which has been pooled and averaged between both stimulation intensities for the TMS session (**E**) and TMES session (**F**), respectively. ^$^ denotes significant difference from all previous power outputs. ^¥^ denotes significant difference from the 25 W condition. ^€^ denotes significant difference from the 25, 50, and 100 W conditions. ^¢^ denotes significant difference from the 25, 50, 100, and 150 W conditions.

**Figure 4 brainsci-09-00205-f004:**
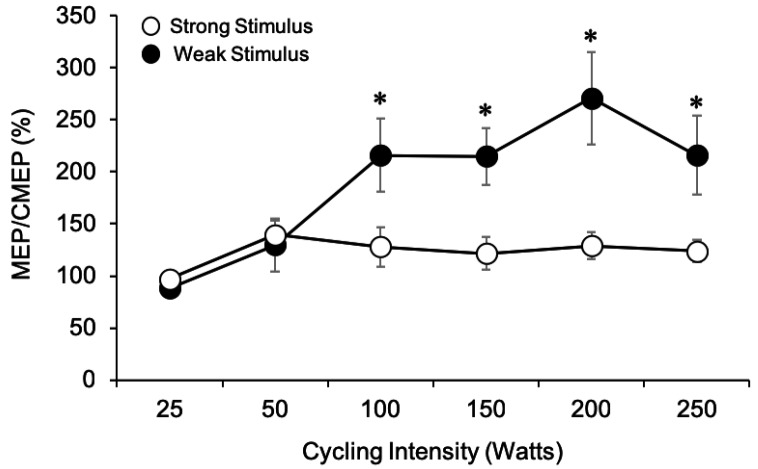
Comparison of MEP/CMEP ratios for the weak (filled circles) and strong (unfilled circles) stimulation intensities as power output increased from 25 W to 250 W. * represents significant difference between stimulation intensities at each given power output (*p* < 0.05). In some cases, SE bars were smaller than the symbols for the data points.

**Figure 5 brainsci-09-00205-f005:**
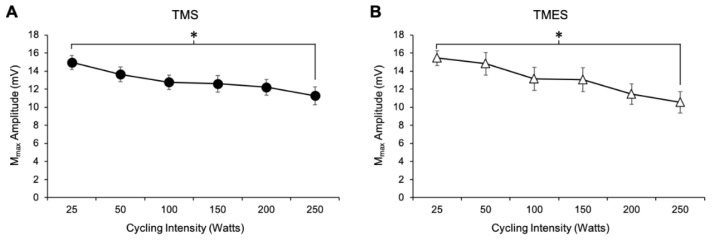
Changes in M_max_ amplitudes with increasing power output pooled between stimulation intensities for the TMS (**A**) and TMES (**B**) session. * denotes significant main effect of power output on M_max_ amplitude. M_max_ decreased by approximately 24.9 and 31.7% as power output increased from 25 to 250 W for the TMS and TMES sessions, respectively.

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
