# Peer review of "Corticospinal-Evoked Responses from the Biceps Brachii during Arm Cycling across Multiple Power Outputs"

_brainsci, 2019, doi:10.3390/brainsci9080205_

Round 1

Reviewer 1 Report

The authors deal with an intriguing and probably under-investigated topic, which is the assessment of corticospinal (MEPs) and cervicomedullary evoked potentials (CMEPs) from an upper limb muscle of healthy subjects during arm cycling and across multiple power outputs. In short, it has been found that MEPs and CMEPs changed in a similar manner and that MEP amplitudes were disproportionately larger than CMEP as power output increased, despite being initially matched in amplitude, particularly with strong stimulation. They conclude that the increase and subsequent plateaus in overall excitability are likely mediated by spinal factors and that supraspinal excitability is enhanced to a larger degree than spinal excitability as the power output of arm cycling increases.

Overall, the study is nicely conceived and designed; the results seem to be consistent and are adequately illustrated and discussed. However, there are different methodological concerns regarding this mansucript.

MAJOR

-  Methods: it should be stated whether some of the participants had taken caffeine, energy-drinks, or any other substance or drug able to affect cortical excitability (Ziemann U. Clin Neurophysiol 2004; Paulus W, et al. Brain Stimul 2008) prior to the enrolment  Additionally, the quality of sleep of the nights preceding the exams should be reported, given that insomnia or sleep deprivation/fragmentation modulate motor cortex excitability and corticospinal conductivity (for a recent comprehensive review, see Lanza G, et al. Sleep Med Rev 2015).

- Methods: why the CMEPs were elicited by transmastoid electrical stimulation and not by transvertebral magnetic stimulation or by means of the F-waves technique, as recommended by the IFCN guidelines (Rossini PM, et al. Clin Neurophysiol 2015)?

- Methods: how randomization was carried out? Were the operators blind to the procedure?

- Methods: as known, the more precise estimation of MEP amplitude is by means of the amplitude ratio (MEP/CMAP) of the motor response (Rossini PM, et al. Clin Neurophysiol 2015). This reliably reflects the excitation status of motor cortical areas and spinal/motor nerve excitability.  

- Methods: the number of TMS meaures included in this study is rather incomplete; for instance, the resting and/or active motor threshold, the contralateral and ipsilateral silent period, and the central motor conduction time are not mentioned. Indeed, they are basic TMS indexes that would have disclosed very useful additional findings in the context of this study.

- Discussion: the limitation paragraph is lacking; in particular, the very small sample size should be acknowledged. Notwithstanding this limitation, a sample of only 6 CMEP is really small (even for TMS studies) and may significantly question the reliability of the statistical analysis.  

- Discussion: although performed in healthy volunteers, the translational findings of this study should be mentioned, for instance: patients affected by typical corticospinal pathologies, such as stroke, chronic cerebrovascular disease, motor neuron disease, spinal cord compression (for a comprehensive review, see Kobayashi and Pascual-Leone. Lancet Neurol 2003); motor deficit due to inter-hemispheric dysconnection (Lanza G, et al. BioMed Res Int 2013); sensory-motor network disorders, such as restless legs syndrome (Lanza G, et al. Ther Adv Neurol Disord 2018). These studies have also intriguing implications in terms of adaptive plasticity of central and peripheral nervous system in response to a stress stimulus of different etiology (Pennisi G, et al. Plos Obne 2014; Bella R, et al. Plos One 2015; Bordet R, et al. BMC Med 2017; Lanza G, et al. Sleep Med 2018).

MINOR

- Abstract: in addition to the aim of the study, the “Background” subsection should also briefly explain the theoretical background and the rationale of the study. In the “Methods” part, please clarify what “weak” and “strong” stimulation intensities stand for.

- Methods: for the sake of completeness, specify whether the subject who did not return on the separate day was the left-handed one.

- Methods: it seems that a monophasic magnetic stimulator was used; if so, please state.

Author Response

We wish to thank each of the reviewers for their suggestions and comments. We sincerely appreciate the time and effort you have invested in our research. We address each concern below.

Reviewer #1 (Comments to the Author):

Comment: The authors deal with an intriguing and probably under-investigated topic, which is the assessment of corticospinal (MEPs) and cervicomedullary evoked potentials (CMEPs) from an upper limb muscle of healthy subjects during arm cycling and across multiple power outputs. In short, it has been found that MEPs and CMEPs changed in a similar manner and that MEP amplitudes were disproportionately larger than CMEP as power output increased, despite being initially matched in amplitude, particularly with strong stimulation. They conclude that the increase and subsequent plateaus in overall excitability are likely mediated by spinal factors and that supraspinal excitability is enhanced to a larger degree than spinal excitability as the power output of arm cycling increases.

Overall, the study is nicely conceived and designed; the results seem to be consistent and are adequately illustrated and discussed. However, there are different methodological concerns regarding this manuscript.

Response: Thank you for your kind words and for taking the time to review our manuscript. We sincerely appreciate the efforts you have put in to enhance our research.

MAJOR

Comment- Methods: it should be stated whether some of the participants had taken caffeine, energy-drinks, or any other substance or drug able to affect cortical excitability (Ziemann U. Clin Neurophysiol 2004; Paulus W, et al. Brain Stimul 2008) prior to the enrolment Additionally, the quality of sleep of the nights preceding the exams should be reported, given that insomnia or sleep deprivation/fragmentation modulate motor cortex excitability and corticospinal conductivity (for a recent comprehensive review, see Lanza G, et al. Sleep Med Rev 2015).

Response: Thank you for this comment. We have added the following sentence in section 2.1 of the manuscript. As for the comment regarding sleep quality, unfortunately this information was not obtained.

L91-92: “Participants also refrained from caffeine for 12 hours and alcohol for 24 hours prior to each experimental session.”

Comment- Methods: why the CMEPs were elicited by transmastoid electrical stimulation and not by transvertebral magnetic stimulation or by means of the F-waves technique, as recommended by the IFCN guidelines (Rossini PM, et al. Clin Neurophysiol 2015)?

Response: In the present study we used TMES to elicit CMEPs as has been done previously in many investigations from our lab (Forman et al. 2018; Forman et al. 2015; 2016a; Forman et al. 2016b; Lockyer et al. 2018; Lockyer et al. 2019; Pearcey et al. 2014; Power et al. 2018; Spence et al. 2016) and that of Gandevia and Taylor. This methodology is the most appropriate, non-invasive, technique currently available to assess motoneurone excitability as discussed in a review paper by the aforementioned authors (McNeil et al. 2013). It is preferable to the F-wave for multiple reasons as discussed in that review and is not possible to elicit in the biceps brachii during arm cycling – we’ve tried!

We did not see reference to TMES-evoked CMEPs in the Rossini paper.

Comment- Methods: how randomization was carried out? Were the operators blind to the procedure?

Response: Randomization of the trials was performed using a random list generator offline prior to data collection. The operators were not blind to the randomization process, however this process was only performed to control for potential order effects from participants performing all trials in a specific order.

Comment- Methods: as known, the more precise estimation of MEP amplitude is by means of the amplitude ratio (MEP/CMAP) of the motor response (Rossini PM, et al. Clin Neurophysiol 2015). This reliably reflects the excitation status of motor cortical areas and spinal/motor nerve excitability. 

Response: MEPs and CMEPs in our study were indeed made relative to the maximal compound muscle action potential (or Mmax) within each trial (see Table 1 and Figures 3A, 3B). I think there may be some confusion regarding terminology. Mmax is another term for CMAP – they both involve stimulation of the peripheral nerve to evoke a potential in the muscle of interest.

Comment- Methods: the number of TMS measures included in this study is rather incomplete; for instance, the resting and/or active motor threshold, the contralateral and ipsilateral silent period, and the central motor conduction time are not mentioned. Indeed, they are basic TMS indexes that would have disclosed very useful additional findings in the context of this study.

Response: Neither resting or active motor threshold were presented in the manuscript because MEPs and CMEPs were initially set to evoke response amplitudes equal to ~10% and ~40% of the maximal compound muscle action potentials. This was done to examine the influence of two different stimulation intensities (weak and strong) on corticospinal and spinal excitability during arm cycling and was indeed one of the purposes of the study. Also, while the reviewer is correct in that other TMS parameters could have been used to potentially yield other information, our primary goal of this research was to assess changes in excitability via changes in response amplitudes from the contracted muscles.

Comment- Discussion: the limitation paragraph is lacking; in particular, the very small sample size should be acknowledged. Notwithstanding this limitation, a sample of only 6 CMEP is really small (even for TMS studies) and may significantly question the reliability of the statistical analysis.

Response: Thank you for this comment. We have added the following to the Methodological Considerations section of the manuscript.

L488-491. “Additionally, the sample size of (n = 9) for MEPs and (n = 8) for CMEPs was not determined by a power analysis and therefore, it is unclear whether a larger sample size would have influenced the present results.”

As for the small sample of 6 MEPs and CMEPs per trial, the small number of stimulations were chosen in efforts to reduce the potential influence of fatigue on our outcome measures. However, with that said recent similar work during locomotor-like movement in the legs (Weavil et al. 2015) has used smaller numbers of stimulations and reported similar findings.

Comment- Discussion: although performed in healthy volunteers, the translational findings of this study should be mentioned, for instance: patients affected by typical corticospinal pathologies, such as stroke, chronic cerebrovascular disease, motor neuron disease, spinal cord compression (for a comprehensive review, see Kobayashi and Pascual-Leone. Lancet Neurol 2003); motor deficit due to inter-hemispheric dysconnection (Lanza G, et al. BioMed Res Int 2013); sensory-motor network disorders, such as restless legs syndrome (Lanza G, et al. Ther Adv Neurol Disord 2018). These studies have also intriguing implications in terms of adaptive plasticity of central and peripheral nervous system in response to a stress stimulus of different etiology (Pennisi G, et al. Plos Obne 2014; Bella R, et al. Plos One 2015; Bordet R, et al. BMC Med 2017; Lanza G, et al. Sleep Med 2018).

Response: We thank the reviewer for this comment and we agree that our work may lead to human applications given that arm cycling given its use in neurorehabilitation. The purpose of the work, however, is basic in nature and as such we do not wish to include this information. The Special Issue to which the paper is submitted also requests non-pathological states.

MINOR

Comment- Abstract: in addition to the aim of the study, the “Background” subsection should also briefly explain the theoretical background and the rationale of the study. In the “Methods” part, please clarify what “weak” and “strong” stimulation intensities stand for.

Response: we appreciate the reviewers comment and normally we include such information. To accommodate to the journal’s word limit guidelines for manuscript abstracts (200 words), however, we were required to leave out this information in order to include what we considered the most relevant information so that readers would understand the methods and finings better. As such we tried to be very direct and succinct as to what exactly we did and why.

Comment- Methods: for the sake of completeness, specify whether the subject who did not return on the separate day was the left-handed one.

Response: The subject who did not return for the second day was not the left-handed participant. The underlined in the following has been added to section 2.1 of the manuscript:

L83. “…eight of those volunteers (1 left-hand dominant) returned…”

Comment- Methods: it seems that a monophasic magnetic stimulator was used; if so, please state.

Response: All transcranial magnetic stimulation was performed with a Magstim 2002 used in single pulse fashion which elicits biphasic MEPs from surface electromyography (see Figure 2A). We are unsure whether the reviewer meant to say monophasic or single pulse.

References

Alcock LR, Spence AJ, Lockyer EJ, Button DC, and Power KE. Short-interval intracortical inhibition to the biceps brachii is present during arm cycling but is not different than a position- and intensity-matched tonic contraction. Exp Brain Res 2019.

Carroll TJ, Zehr EP, and Collins DF. Modulation of cutaneous reflexes in human upper limb muscles during arm cycling is independent of activity in the contralateral arm. Exp Brain Res 161: 133-144, 2005.

Collins BW, Pearcey GEP, Buckle NCM, Power KE, and Button DC. Neuromuscular fatigue during repeated sprint exercise: underlying physiology and methodological considerations. Appl Physiol Nutr Me 43: 1166-1175, 2018.

Donges SC, Taylor JL, and Nuzzo JL. Elbow angle modulates corticospinal excitability to the resting biceps brachii at both spinal and supraspinal levels. Exp Physiol 104: 546-555, 2019.

Forman DA, Monks M, and Power KE. Corticospinal excitability, assessed through stimulus response curves, is phase-, task-, and muscle-dependent during arm cycling. Neurosci Lett 2018.

Forman DA, Philpott DT, Button DC, and Power KE. Cadence-dependent changes in corticospinal excitability of the biceps brachii during arm cycling. J Neurophysiol 114: 2285-2294, 2015.

Forman DA, Philpott DT, Button DC, and Power KE. Differences in corticospinal excitability to the biceps brachii between arm cycling and tonic contraction are not evident at the immediate onset of movement. Exp Brain Res 234: 2339-2349, 2016a.

Forman DA, Richards M, Forman GN, Holmes MW, and Power KE. Changes in Corticospinal and Spinal Excitability to the Biceps Brachii with a Neutral vs. Pronated Handgrip Position Differ between Arm Cycling and Tonic Elbow Flexion. Front Hum Neurosci 10: 543, 2016b.

Kennefick M, Burma JS, van Donkelaar P, and McNeil CJ. The Time Course of Motoneuronal Excitability during the Preparation of Complex Movements. J Cognitive Neurosci 31: 781-790, 2019.

Lockyer EJ, Benson RJ, Hynes AP, Alcock LR, Spence AJ, Button DC, and Power KE. Intensity matters: effects of cadence and power output on corticospinal excitability during arm cycling are phase- and muscle-dependent. J Neurophysiol 2018.

Lockyer EJ, Nippard AP, Kean K, Hollohan N, Button DC, and Power KE. Corticospinal Excitability to the Biceps Brachii is Not Different When Arm Cycling at a Self-Selected or Fixed Cadence. Brain Sci 9: 2019.

Martin PG, Gandevia SC, and Taylor JL. Output of human motoneuron pools to corticospinal inputs during voluntary contractions. J Neurophysiol 95: 3512-3518, 2006.

McNeil CJ, Butler JE, Taylor JL, and Gandevia SC. Testing the excitability of human motoneurons. Front Hum Neurosci 7: 152, 2013.

Pearcey GE, Power KE, and Button DC. Differences in Supraspinal and Spinal Excitability during Various Force Outputs of the Biceps Brachii in Chronic- and Non-Resistance Trained Individuals. Plos One 9: e98468, 2014.

Power KE, Lockyer EJ, Forman DA, and Button DC. Modulation of motoneurone excitability during rhythmic motor outputs. Appl Physiol Nutr Metab 1-10, 2018.

Spence AJ, Alcock LR, Lockyer EJ, Button DC, and Power KE. Phase- and Workload-Dependent Changes in Corticospinal Excitability to the Biceps and Triceps Brachii during Arm Cycling. Brain Sci 6: 2016.

Taylor JL. Stimulation at the cervicomedullary junction in human subjects. J Electromyogr Kinesiol 16: 215-223, 2006.

Weavil JC, Sidhu SK, Mangum TS, Richardson RS, and Amann M. Intensity-dependent alterations in the excitability of cortical and spinal projections to the knee extensors during isometric and locomotor exercise. Am J Physiol Regul Integr Comp Physiol 308: R998-1007, 2015.

Zehr EP, and Chua R. Modulation of human cutaneous reflexes during rhythmic cyclical arm movement. Exp Brain Res 135: 241-250, 2000.

Zehr EP, and Duysens J. Regulation of arm and leg movement during human locomotion. Neuroscientist 10: 347-361, 2004.

Reviewer 2 Report

The manuscript points to important methodological and physiological aspects about excitability of muscles at different outputs of contraction. I do see the relevance and I do see the rigorousness of methodology that sustains the findings. Therefore I would like to recommend publication of the article, however, after major revision.

Some points lack detail of description and the readability of manuscript, sometimes lacks of continuity

1.- Line 116. Usually ground electrode is placed between stimulus and recording site. So, why in this case is placed distally to biceps brachialis at lateral epicondyle?. Can this placement affect the baseline and, then, eventually affect to the amplitude of Mmax electrically elicited?. See the bottom row of figure 2. Please, clarify this point.

2.- Line 196. Authors say that “1 Mmax, 6 MEPs and 6 CMEPs were evoked”. However at line 187 they state that “participants were randomly assigned to complete either session one (TMS) or session two (TMES) first”, and in line 242 they say “MEPs and CMEPs (…) were evoked on separate days”. So, I understand that protocol included 1 Mmax and either 6 MEPs or 6 CMEPs, but not both.

3.- Line 212. Authors describe that “pre-stimulus EMG was measured from the rectified virtual channel created for the biceps and triceps brachii”. This methodological aspect is very relevant. What exactly was measured?, because you can measure several variables from an EMG recording, and  besides, what is a rectified virtual channel?. From this sentence it seems to me that only one (virtual) channel is determined, however in figure 3C, D I see graphs for biceps brachialis pre-stim and in 3E,F for triceps.

4.- Authors have used amplitude of evoked potentials to assess excitability. However, it is well-kwon the of pseudofacilitation effect (increase in amplitude with corresponding reduction in duration of the negative phase of the M wave, resulting in no change in area). Then, how do they discard this effect and, in close relationship with this, why they do not used the area instead of amplitude to assess excitability?. Authors discuss the results of literature obtained with area measurements (Martin et al.) but, can be compared both kind of measures?

5.- Table 1 show the same results that are plotted in figures, therefore, this information is redundant and should be removed.

6.- Along the results, the MEP and CMEP are cited as the main measurement (e.g, lines 253-259 or 296-298), however, they are indicated as %Mmax in figure 3. However, both measurements are not the same, because the decrease in Mmax (the denominator of the quotient) with output intensity implies that MEP/CMEP can increase without a real change in amplitude. Please show that increase is not due to this effect.

7.- Figure 2. I have mentioned above the oscillation in baseline at the bottom row of the figure. Besides, I find difficult to explain the clear increase in latency between stimulus artefact and the onset of Mmax compared with the rest of waveforms.

8.- There is a clear loss of continuity when authors finish the exposure of results for figures 4 and 5 and suddenly they go back to figure 3 for Pre-stimulus EMG. Please group text and figures.

9.- I have not find a clear explanation in Discussion of the decrease of Mmax with increasing of output power and this is for me a very  intriguing fact. Please explain in more detail.

Author Response

We wish to thank each of the reviewers for their suggestions and comments. We sincerely appreciate the time and effort you have invested in our research. We address each concern below.

Reviewer #2 (Comments to the Author):

Comment: The manuscript points to important methodological and physiological aspects about excitability of muscles at different outputs of contraction. I do see the relevance and I do see the rigorousness of methodology that sustains the findings. Therefore, I would like to recommend publication of the article, however, after major revision.

Some points lack detail of description and the readability of manuscript, sometimes lacks of continuity

Response: Thank you for taking the time to review our manuscript and for these insightful comments.

Comment: 1.- Line 116. Usually ground electrode is placed between stimulus and recording site. So, why in this case is placed distally to biceps brachialis at lateral epicondyle? Can this placement affect the baseline and, then, eventually affect to the amplitude of Mmax electrically elicited? See the bottom row of figure 2. Please, clarify this point.

Response: When assessing the biceps brachii it is common place for the ground electrode to be placed on an area of low electrical activity, such as a bony prominence. Using the differential amplifiers that we did, this ensures that the two recording electrodes compare their signals to the ground and common signal is between those recording electrodes is then removed as noise. The current placement of electrodes will not negatively impact the recordings from the biceps brachii.

Comment: 2.- Line 196. Authors say that “1 Mmax, 6 MEPs and 6 CMEPs were evoked”. However at line 187 they state that “participants were randomly assigned to complete either session one (TMS) or session two (TMES) first”, and in line 242 they say “MEPs and CMEPs (…) were evoked on separate days”. So, I understand that protocol included 1 Mmax and either 6 MEPs or 6 CMEPs, but not both.

Response: Thank you for this comment. The text has been altered and now reads as follows (changes are underlined):

L198-199: “… 6 o’clock position, 1 Mmax and either 6 MEPs or 6 CMEPs (depending on the session) were evoked in a randomized order.”

Comment: 3.- Line 212. Authors describe that “pre-stimulus EMG was measured from the rectified virtual channel created for the biceps and triceps brachii”. This methodological aspect is very relevant. What exactly was measured?, because you can measure several variables from an EMG recording, and besides, what is a rectified virtual channel?. From this sentence it seems to me that only one (virtual) channel is determined, however in figure 3C, D I see graphs for biceps brachialis pre-stim and in 3E,F for triceps.

Response: Thank you for this comment. We created separate rectified virtual channels for both the biceps brachii and the triceps brachii and then proceeded to take the mean value of EMG from a period 50 ms prior to each stimulation. This provides us with a measure of background or pre-stimulus EMG for each muscle and stimulation type as previously completed in our lab and by others (Alcock et al. 2019; Collins et al. 2018; Forman et al. 2018; Forman et al. 2015; 2016a; Forman et al. 2016b; Lockyer et al. 2018; Lockyer et al. 2019; Spence et al. 2016) (Carroll et al. 2005; Zehr and Chua 2000; Zehr and Duysens 2004).

Comment: 4.- Authors have used amplitude of evoked potentials to assess excitability. However, it is well-kwon the of pseudofacilitation effect (increase in amplitude with corresponding reduction in duration of the negative phase of the M wave, resulting in no change in area). Then, how do they discard this effect and, in close relationship with this, why they do not used the area instead of amplitude to assess excitability? Authors discuss the results of literature obtained with area measurements (Martin et al.) but, can be compared both kind of measures?

Response: The reviewer has raised an issue that we have thought about and assessed multiple times in prior works. We have chosen to use amplitude as our measure given that all previous work from our lab has done the same because we do not see any differences in the results based on using area or amplitude. Amplitude is our preferred method because it is more easily assessed an intuitive for the readers as opposed to areas. Our understanding is that area is typically used in studies that either involve fatigue or set the backdrop for follow-up studies using fatigue.

Comment: 5.- Table 1 show the same results that are plotted in figures, therefore, this information is redundant and should be removed.

Response: Thank you for this comment. Table 1 and its citations have been removed. 

Comment: 6.- Along the results, the MEP and CMEP are cited as the main measurement (e.g, lines 253-259 or 296-298), however, they are indicated as %Mmax in figure 3. However, both measurements are not the same, because the decrease in Mmax (the denominator of the quotient) with output intensity implies that MEP/CMEP can increase without a real change in amplitude. Please show that increase is not due to this effect.

Response: MEP involves the brain, spinal cord and nerve muscle. CMEPs involve spinal cord and nerve muscle. Mmax involves nerve muscle. Thus, when MEPs and CMEPs are made relative to Mmax, the common ‘nerve muscle’ response is accounted for and you are left with brain and spinal and spinal only excitability, respectively. Using the same reasoning, when MEP is made relative to CMEP, they both involve the nerve muscle response. As such, the effect on the MEP and CMEP will be the same, because the same Mmax is used. In addition, both have spinal excitability accounted for, which leaves the MEP/CMEP response as a measure of brain/cortical excitability.

We assure the reviewer that this method is commonly used for the above reasons to give an indication of cortical excitability (Donges et al. 2019; Kennefick et al. 2019; Taylor 2006).

Comment: 7.- Figure 2. I have mentioned above the oscillation in baseline at the bottom row of the figure. Besides, I find difficult to explain the clear increase in latency between stimulus artefact and the onset of Mmax compared with the rest of waveforms.

Response: Thank you for this comment. The oscillation in Mmax throughout the experiment is similar to what is shown in Figure 4A of Martin et al., (2006) although not specifically discussed in that paper. The latency of the Mmax did not change throughout the experiment, the final waveform included in this figure is a mistake (actually was a MEP) and has been corrected.

Comment: 8.- There is a clear loss of continuity when authors finish the exposure of results for figures 4 and 5 and suddenly they go back to figure 3 for Pre-stimulus EMG. Please group text and figures.

Response: The authors agree and thank the reviewer for this comment. We have now placed the information on bEMG within the sections on MEP and CMEP for clarity.

Comment: 9.- I have not find a clear explanation in Discussion of the decrease of Mmax with increasing of output power and this is for me a very intriguing fact. Please explain in more detail.

Response: We haven’t included a discussion on this topic because in all likelihood it is not mechanistic and doesn’t impact the findings regarding corticospinal excitability which was the objective of the present study. When the participants cycle at higher workloads they voluntarily recruit higher numbers of spinal motoneurones. Because the nerve stimulation excites the axons of the motoneurones, as power output increases the ability of the nerve stimulation to excite the axons will be reduced because they are already active (voluntarily). In other words, you can’t activate the motor axons once they are already active. Having said that, nerve stimulation will always produce a Mwave because the stimulation can still insert an extra ‘pulse’ to enhance axon firing via summation and also because the power outputs used were not truly maximal, but the amplitudes will still be reduced. This effect has also been shown during isometric contractions (Martin et al. 2006).

References

Alcock LR, Spence AJ, Lockyer EJ, Button DC, and Power KE. Short-interval intracortical inhibition to the biceps brachii is present during arm cycling but is not different than a position- and intensity-matched tonic contraction. Exp Brain Res 2019.

Carroll TJ, Zehr EP, and Collins DF. Modulation of cutaneous reflexes in human upper limb muscles during arm cycling is independent of activity in the contralateral arm. Exp Brain Res 161: 133-144, 2005.

Collins BW, Pearcey GEP, Buckle NCM, Power KE, and Button DC. Neuromuscular fatigue during repeated sprint exercise: underlying physiology and methodological considerations. Appl Physiol Nutr Me 43: 1166-1175, 2018.

Donges SC, Taylor JL, and Nuzzo JL. Elbow angle modulates corticospinal excitability to the resting biceps brachii at both spinal and supraspinal levels. Exp Physiol 104: 546-555, 2019.

Forman DA, Monks M, and Power KE. Corticospinal excitability, assessed through stimulus response curves, is phase-, task-, and muscle-dependent during arm cycling. Neurosci Lett 2018.

Forman DA, Philpott DT, Button DC, and Power KE. Cadence-dependent changes in corticospinal excitability of the biceps brachii during arm cycling. J Neurophysiol 114: 2285-2294, 2015.

Forman DA, Philpott DT, Button DC, and Power KE. Differences in corticospinal excitability to the biceps brachii between arm cycling and tonic contraction are not evident at the immediate onset of movement. Exp Brain Res 234: 2339-2349, 2016a.

Forman DA, Richards M, Forman GN, Holmes MW, and Power KE. Changes in Corticospinal and Spinal Excitability to the Biceps Brachii with a Neutral vs. Pronated Handgrip Position Differ between Arm Cycling and Tonic Elbow Flexion. Front Hum Neurosci 10: 543, 2016b.

Kennefick M, Burma JS, van Donkelaar P, and McNeil CJ. The Time Course of Motoneuronal Excitability during the Preparation of Complex Movements. J Cognitive Neurosci 31: 781-790, 2019.

Lockyer EJ, Benson RJ, Hynes AP, Alcock LR, Spence AJ, Button DC, and Power KE. Intensity matters: effects of cadence and power output on corticospinal excitability during arm cycling are phase- and muscle-dependent. J Neurophysiol 2018.

Lockyer EJ, Nippard AP, Kean K, Hollohan N, Button DC, and Power KE. Corticospinal Excitability to the Biceps Brachii is Not Different When Arm Cycling at a Self-Selected or Fixed Cadence. Brain Sci 9: 2019.

Martin PG, Gandevia SC, and Taylor JL. Output of human motoneuron pools to corticospinal inputs during voluntary contractions. J Neurophysiol 95: 3512-3518, 2006.

McNeil CJ, Butler JE, Taylor JL, and Gandevia SC. Testing the excitability of human motoneurons. Front Hum Neurosci 7: 152, 2013.

Pearcey GE, Power KE, and Button DC. Differences in Supraspinal and Spinal Excitability during Various Force Outputs of the Biceps Brachii in Chronic- and Non-Resistance Trained Individuals. Plos One 9: e98468, 2014.

Power KE, Lockyer EJ, Forman DA, and Button DC. Modulation of motoneurone excitability during rhythmic motor outputs. Appl Physiol Nutr Metab 1-10, 2018.

Spence AJ, Alcock LR, Lockyer EJ, Button DC, and Power KE. Phase- and Workload-Dependent Changes in Corticospinal Excitability to the Biceps and Triceps Brachii during Arm Cycling. Brain Sci 6: 2016.

Taylor JL. Stimulation at the cervicomedullary junction in human subjects. J Electromyogr Kinesiol 16: 215-223, 2006.

Weavil JC, Sidhu SK, Mangum TS, Richardson RS, and Amann M. Intensity-dependent alterations in the excitability of cortical and spinal projections to the knee extensors during isometric and locomotor exercise. Am J Physiol Regul Integr Comp Physiol 308: R998-1007, 2015.

Zehr EP, and Chua R. Modulation of human cutaneous reflexes during rhythmic cyclical arm movement. Exp Brain Res 135: 241-250, 2000.

Zehr EP, and Duysens J. Regulation of arm and leg movement during human locomotion. Neuroscientist 10: 347-361, 2004.

Round 2

Reviewer 1 Report

The authors have adequately addressed my concerns, thus improving the quality of this manuscript. I do not have further comments.

Reviewer 2 Report

Authors have satisfactorily addressed all the questions rose. I have no more significant queries